# Leadership Succession Planning for Professional Nurses in a Selected Public Hospital in Mangaung District, Free State Province: A Study Protocol

**DOI:** 10.3390/healthcare13243313

**Published:** 2025-12-18

**Authors:** Lebogang G. Motlhaole, Aluwani D. Mudzweda, Takalani R. Luhalima

**Affiliations:** Department of Advanced Nursing Sciences, Faculty of Health Sciences, University of Venda, Thohoyandou 0950, South Africa; 24049746@mvula.univen.ac.za

**Keywords:** leadership, nurse managers, professional nurses, public hospitals, succession planning

## Abstract

Lack of leadership succession planning in South African public hospitals places nursing leadership at great risk instead of improving healthcare. There is a significant demand for nurse managers in the Free State Province; therefore, leadership succession planning is important. The re-advertising of unfilled leadership roles, the projected volume of nurse managers who will be retiring, and the number of professional nurses opting for better international opportunities indicate the need for effective succession planning. The study aims to determine leadership succession planning for professional nurses in a selected public hospital in the Mangaung District, Free State Province. A qualitative, explorative, and descriptive research design will be used. Non-probability purposive sampling will be adopted to explore the leadership succession planning. The research participants will consist of professional nurses who are currently permanently employed within the Mangaung district, Free State Province. The sample size will be determined by data saturation. An estimated sample size of ±20 participants will be expected. Data collection will be performed through in-depth, unstructured interviews to answer the research question. A central place for interviews will be organised, and appointments will be made with participants as per their schedule or availability. Data will be analysed using Braun and Clarke’s thematic method. The conclusion and the recommendations will be based on the findings of the study.

## 1. Introduction

Succession planning is regarded as a deliberate approach to maintaining the smooth functioning of an organisation despite the increasing management turnover ref. [1]. It is a deliberate approach created to focus on nurturing prospective managers and emphasising leadership flow without disruptions ref. [2,3]. Succession planning is a continuous cycle whereby the organisation identifies prospective managers and prepares them to take on management positions should they arise. Globally, factors such as a lack of skilled human resources and limited time frames to find suitable candidates are among the factors that have given rise to succession planning in some institutions ref. [4]. Many institutions are implementing succession planning as a strategy to develop and retain their workforce, facilitating a smoother selection process ref. [5]. In a study conducted by ref. [6] in Iran, it was found that some of the reasons why there is no successful leadership succession planning in place are due to the failure of not being able to understand the term succession planning, staff not accepting new management, and there being no set criteria for selection. Furthermore, the results indicated that fear of losing the experience and skills of nurses, a halt on organisational programmes, and a lack of leadership also contributed. Additionally, some managers are of the opinion that their successors will assume their roles before they retire. In Brazil, ref. [7] indicated that the lack of suitable candidates to assume nurse leadership positions was due to a shortage of nurses, resulting in the healthcare facilities’ failure to thrive.

It has been noted that Saudi Arabia has subsequently seen an increase in its population, resulting in a shortage of nurses ref. [8]. A study conducted in Saudi Arabia by ref. [9] alluded that nurses lack the necessary leadership skills to mitigate the ever-changing healthcare system that requires a multifaceted approach to patients’ well-being. Additionally, many nurse manager positions are held by foreign nationals, thus indicating that the country is very dependent on foreigners to run its healthcare institutions ref. [9]. However, ref. [9] further concluded that the lack of plans in the development of professional nurses and in nursing education are some of the factors that lead to the ineffective implementation of leadership succession planning.

Ref. [10] noted that the Registered Nurses and Midwives Association in Ghana rolled out a framework for the years 2024–2028 on 8 August 2024, whereby newly appointed nurses must be orientated and trained into leadership positions. In addition, leadership development programmes are to be established for all senior nurses as well as mentorship programmes. The Nursing and Midwifery Council of Nigeria has initiated leadership development programmes as a way of enhancing skills such as management and leadership amongst young nurses in Nigeria ref. [11]. In Kenya, as per The Ministry of Health’s National Nursing and Midwifery Policy (2022–2032), recommendations were made regarding strengthening management and leadership practices, emphasising the importance of nursing leadership and the need for the improvement and development of nurse leaders ref. [12].

Most public hospitals within South Africa (SA) are faced with an ageing nurse manager population. Therefore, it is imperative that leadership succession planning be adopted to mitigate this phenomenon before it is too late. According to the statistics as observed by the South African Nursing Council (SANC) in 2022, 52,776 professional nurses are close to retirement within the next 10 years or so, and most of these professional nurses are nurse managers from the age of 55 and up ref. [13]. It is in this regard that the researcher aims to address a vast gap identified when it comes to SA’s succession planning, particularly for nurses. Succession planning is practiced more in private institutions as compared to public ones ref. [14]. It has been indicated that 47% of nurses are expected to retire within the next 15 years ref. [15]. Moreover, an estimated number of 34,000 registered nurses will be needed by 2025, as reported in SA’s 2030 Human Resources for Health Strategy by ref. [16]. Factors that hinder leadership succession planning in the SA state-owned institutions include a lack of support from the organisation, managerial supervision, and the need for performance appraisal ref. [17].

Since most professional nurses are on the verge of retirement, there might be a gap in leadership positions being occupied in hospitals. Consequently, and for the DoH to be able to sustain human resources, fulfil S.A. citizens’ health needs, and attain adequate healthcare services, leadership succession planning is recommended ref. [18]. Ref. [18] further indicated that few nurses occupy higher-ranking leadership positions, such as clinical heads, and barriers such as failure to acknowledge postgraduate qualifications, a lack of succession planning, and a lack of mentoring are in place. The continued success of the healthcare facilities performing their best, delivering quality and safe nursing care, lies in the up-skilling and retention of their current human resources. The study aims to determine leadership succession planning for professional nurses in a selected public hospital in the Mangaung District, Free State Province.

### 1.1. Theoretical Framework

This study focuses on transformational leadership theory, which emphasises leaders recognising and supporting the well-being of their subordinates. Ref. [19] notes that the theory aims to inspire followers to emulate their leaders, helping them align personal aspirations with organisational goals, fostering commitment and loyalty to the organization among followers. The theory outlines four key components as indicated by Ali Azimi [20]: Idealised Influence (II), where transformational leaders mentor future nurse leaders; Inspirational Motivation (IM), focusing on continuity through effective succession planning; Intellectual Stimulation (IS), promoting a culture of growth and innovation within healthcare facilities; and Individualised Consideration (IC), ensuring employee involvement and motivation. Together, these elements underscore the importance of leadership succession planning in maintaining organisational effectiveness and fostering employee engagement. Transformational leadership theory was selected to determine leadership succession planning in the Free State public hospital, where there is a decline in nurse managers primarily due to the retirement of personnel, posing a risk to patient care, management, and nursing staff retention.

### 1.2. Problem Statement

Researchers have observed from the recruitment page of the Free State Department of Health (FSDoH) that, in 2024 alone, close to 20 nurse manager posts were advertised within Mangaung, Free State (FS) Province. The shortage of nurse managers has resulted in an increase in workload, as managers are often responsible for two units. According to ref. [21], 1435 professionals in the Free State Province are expected to retire in the next 5–10 years. Moreover, succession planning is needed so that leaders can address the existing challenges of an ageing workforce that is close to retirement; the limited healthcare resources, equipment, and personnel; and the need for a sustainable infrastructure.

The failure to retain suitable nursing managers is attributed to the lack of leadership succession planning, as observed by the re-advertising of certain positions. Furthermore, ref. [22] indicates that SA requires 391 nursing managers and 9005 professional nurses. With limited mentors being available, vacant leadership roles will not be filled sooner due to a lack of experienced personnel. Furthermore, grooming future nurse leaders remains a challenge due to limited development opportunities. Lack of, or poor, leadership succession planning may result in medical errors and patient mortality ref. [23,24]. Challenges related to failure to implement leadership succession planning are directly linked to poor management, a lack of development of nurse managers, a lack of policies and set criteria of who is eligible for succession planning, and a lack of resources ref. [6,24,25]. Limited research has been conducted to understand the importance of leadership succession planning for public hospitals. The study aims to determine the leadership succession planning for professional nurses in a selected public hospital in the Mangaung District, Free State Province.

### 1.3. Rationale of the Study

The study addresses the leadership gap in nursing, which is exacerbated by the increasing complexities in healthcare, and highlights the critical role of professional nurses in leadership succession. Despite the importance of this progression, it is insufficiently supported and understood. A literature review revealed a lack of research on leadership succession in public hospitals, particularly in the context of nursing. Furthermore, unclear promotion pathways and insufficient mentorship impede career progression and leadership development ref. [18]. The South African Nursing Leadership Initiative and various pilot programmes have recognised the necessity for organised leadership development; however, the execution of these initiatives varies significantly among provinces.

There is a lack of established formal policies and procedures in public hospitals, which are essential in identifying, mentoring, and preparing future nurse leaders. Additionally, integrating leadership development into routine operations should include incentives, training, and strategic planning. The relevance of this study will provide meaningful insights that can influence policymakers, address a critical workforce sustainability issue in SA, and improve patient care and staff retention. The study aims to explore leadership succession planning for professional nurses in a selected public hospital in the Mangaung District, Free State Province.

## 2. Materials and Methods

### 2.1. Objectives

The protocol objectives are as follows:Explore and describe the current state of leadership succession planning.Suggest recommendations for leadership succession planning.

### 2.2. Research Methodology

A qualitative approach will be employed to determine leadership succession planning. This approach enables the researcher to understand the real-life experiences of professional nurses regarding leadership succession planning ref. [26]. Qualitative research allows participants to express their in-depth opinions regarding leadership succession planning ref. [27]. Through in-depth interviews, the study will explore and describe leadership succession planning. Participants will be encouraged to articulate in their own words the significance of leadership succession planning in this public hospital ref. [26].

#### 2.2.1. Study Design

##### Explorative and Descriptive Design

This study employs a sequential methodological approach, beginning with an exploratory design and transitioning to a descriptive design, to determine leadership succession planning among professional nurses in the Mangaung District, Free State. The initial exploratory phase is critical for assessing the feasibility of the study, refining the research question, and establishing a broad conceptual understanding of the current state of leadership succession ref. [28]. This phase will yield preliminary data essential for identifying the core problem statement, exploring the importance of succession planning, and formulating initial recommendations for its implementation. The findings from this exploration will subsequently inform and necessitate the descriptive phase, whose primary objective is to accurately and thoroughly describe the importance of effective leadership succession planning as perceived by participants ref. [28]. By gathering rich, descriptive data from professional nurses about the significance of succession plans in their public hospitals, the descriptive design aims to provide a deeper understanding of the phenomenon, addressing the core questions of what and where the problem lies, thus generating meaningful discoveries for practice and future academic inquiry.

#### 2.2.2. Study Setting

The study will be conducted in the Mangaung District, Free State province, in South Africa. The study will be conducted at the selected public hospital in the Mangaung district, situated in Heidedal, south Bloemfontein, Free State Province, South Africa. The district is home to 3 district hospitals, 1 regional hospital, 1 tertiary hospital, 1 specialised hospital, and a psychiatric complex. The selected public hospital has a total of 720 beds with multiple specialties being offered, such as medicine, obstetrics, orthopaedics, paediatrics, surgery, and trauma units, which are under the leadership of nurses. There are 4 nursing managers, 5 assistant managers, and 33 operational managers, some of whom hold specialised postgraduate qualifications. This facility is one of the two specialised referral hospitals within the Free State region. The team consists of professionals with strong educational backgrounds and formal qualifications. It is diverse, with personnel aged between 20 and 64 years, including both males and females from various races and ethnicities.

#### 2.2.3. Study Population

In this study, the study population is all professional nurses who are working in the public hospital of Mangaung District, Free State Province. The target population will comprise professional nurses who are viewed as potential leaders or yearn to be leaders and have the necessary experience, leadership skills, and education to assume leadership positions. The accessible population will include professional nurses who are still employed in selected facilities, ranging from novice to an expert within the nursing profession, and have the desire to be nurse managers and demonstrate leadership potential.

#### 2.2.4. Sampling

Sampling will be performed in multiple stages. There will be a sampling of facilities and a sampling of participants.

##### Sampling of the Facility

In this protocol, the facilities where the study will be conducted will be sampled using a non-probability purposive sampling. The facilities will be selected based on the observed repetition of nurse manager posts being advertised. Furthermore, these facilities have an ageing group of nurse managers who are due to retire within the next 5–10 years, as compared to other healthcare institutions within the Mangaung district. The selected public hospital is based in the centre of the Free State, enabling a vast attraction of participants, and it is known as a reputable public hospital throughout the Free State.

##### Sampling of Participants

A non-probability purposive sampling will be used to identify professional nurses in the selected healthcare facilities. A sample will be drawn from professional nurses in the selected public hospital. These individuals, the researchers, are of sound mind; they possess the necessary skills and knowledge to fully answer the research question.

##### Sample Size

In this study, the estimated number will be ±20 participants who are currently permanently employed in the Mangaung district with 5 years or more clinical experience. However, the sample size in qualitative research will be determined by data saturation.

#### 2.2.5. Inclusion and Exclusion Criteria

##### Inclusion Criteria

Participants will be registered professional nurses with the South African Nursing Council (SANC) and in possession of updated annual practicing certificates. They must be assuming leadership roles within their respective units/departments, such as being shift leaders. A senior registered professional nurse with 5 years or more clinical experience must be permanently employed in one of the selected healthcare facilities.

##### Exclusion Criteria

Registered professional nurses with less than 5 years of clinical experience and locum registered professional nurses with less than 5 years of clinical experience will be excluded. They are not eligible for managerial roles as the minimum requirements are 5 years or more of clinical experience, and at times, that experience should not be interrupted.

#### 2.2.6. Pre-Testing

Pre-testing will allow the interviewer to anticipate potential problems, why they occur, and how to prevent them when conducting the main study. Above all, it creates an opportunity for the interviewer to hone their interviewing skills. Pre-testing will include 3 participants from the study population. If there are identified problems, these will be resolved before the main study is conducted. Furthermore, if the selected participants do not answer the research question as expected, the research question will be restructured to best suit the study and the understanding of the participants. In cases where the pre-test is a success and no weaknesses of the study are found, the data will be included in the main study. If objectives are not met, the results will be excluded from the study.

#### 2.2.7. Trustworthiness

Ref. [27] states that trustworthiness is a way the researcher presents his/her findings truthfully and honestly to maintain the integrity of the study. Ref. [29] indicated that trustworthiness is critical in authenticating the research outcomes and includes credibility, dependability, transferability, and confirmability.

##### Credibility

Credibility refers to confidence in the truth of the data and its interpretations ref. [30]. For this study, credibility will be enhanced through peer debriefing with experienced researchers, member checking with selected participants to review and validate the interpretation of their inputs, and rigorous evaluation of references based on author expertise, publication source, and relevance to nursing leadership.

##### Dependability

Dependability is performed to illustrate that the findings will be the same if the same group of participants and data analysts are used for a study ref. [31]. A central question will be used to explore participants’ experiences with leadership succession planning. This will ensure uniformity while allowing flexibility for rich data. Detailed documentation of all research steps, including data collection procedures and consultation with academic supervisors to review interpretations and challenge potential bias.

##### Transferability

By explicitly being thorough and giving more details in terms of the findings, this allows readers to better relate to the findings and current situations they will be in, thus ensuring the transferability of the study ref. [32]. In this study, transferability will be achieved through a clear and detailed description of the research study setting, and the sample size will be outlined.

##### Confirmability

Confirmability is a way of ensuring the findings are fair and are a true reflection of the participants ref. [31]. The researcher will not be biased throughout the interviewing stage and will not influence any participants. The researcher will engage in bracketing, i.e., documenting personal assumptions and setting them aside throughout the research process. This will ensure that participants’ voices are authentically represented and that findings are grounded in their lived experiences rather than researcher expectations. The researcher will also utilise an audit trail to jot down topics of interest/discussions, how themes were coded, and provide rationales thereof.

#### 2.2.8. Data Collection

Data will be collected through unstructured interviews with professional nurses who meet the inclusion criteria. Unstructured interviews are effective for obtaining raw data, are easy to analyse, and allow the interviewer to probe for more in-depth details ref. [27]. The following central question will be asked: “How do you view leadership succession planning in your hospital, and what challenges or opportunities do you see in implementing it?”. Unstructured interviews resemble a conversation in face-to-face individual interviews, which encourages participants to engage more openly about leadership succession planning. By conducting these interviews, the researcher can observe both verbal and non-verbal cues from the participants. Additionally, this method reduces anxiety levels for both participants and novice researchers. The researchers’ skills in conducting interviews will facilitate the gathering of rich, raw data and provide descriptive answers that will be valuable for data collection. Moreover, this will help participants feel at ease, encouraging them to share more information essential to their study.

The interview sessions will be recorded for accuracy, and participants will be asked to sign an informed consent form to consent to the recordings. The language of the medium will be English, as it is assumed that all participants are orientated to the language. An office space will be used to conduct interviews within the selected healthcare facilities. A duration of 2–3 months will be suitable to obtain the assumed data. Participants are expected to share their lived experiences as they occur throughout their professional careers. Interviews are expected to last for about 45 min and will be conducted during the participants’ allocated lunch times so as not to disturb the daily activities of the ward.

##### Recruitment of Participants

Recruitment: Firstly, permission will be obtained from the University of Venda Higher Degrees Committee and clearance from the Faculty of Health Science Research Ethics Committee. An appointment will be set up via email detailing the purpose of the study to the Free State Head of Department to obtain permission and consent to conduct the study. After obtaining permission from the FSDoH, a meeting will be set up with the Chief Executive Officer (CEO) of the healthcare facilities to obtain permission and consent. The researcher will use the nursing manager and the human resource department to find suitable candidates to participate in the study. Prospective candidates will be contacted, and the study will be outlined for them. An appointment will be made with the senior professional nurses to outline the purpose of the study, and expectations will be cleared from both the researcher and participants. Those who agree to participate will be issued a consent form to sign. Voluntary participation will be encouraged for the participants.

##### Researcher as an Instrument

In this study, the researchers will serve as instruments by conducting interviews while observing participants’ interactions and behaviours. This approach enables the researcher to describe participants’ reactions to the research questions. The researcher will prevent bias through peer debriefing by engaging with experts and peers in the field about the study’s outcome, using a diary/journal to write down their own judgments, expectations, and experiences of the whole study, and encouraging participants to detail their understanding well without hesitation.

##### Plan for Data Management

The researchers will keep the collected data backed up on iCloud (windows, version 15.0.214) and only the researchers will have access to it, as it will be password-protected. Any data required by the research supervisor will then be sent through an encrypted PDF format. Furthermore, any notes made during data collection will be kept in a lockable cupboard and secured for a minimum of 5 years.

#### 2.2.9. Data Analysis

Data analysis is performed simultaneously with data collection in qualitative research. Data for this study will be collected through audio recordings and interviews. However, it is essential to verify that the collected information relates directly to the research questions. The researcher will review the recordings and make notes during data collection to understand the participants’ perspectives on leadership succession planning.

For data analysis, the researcher will employ Braun and Clarke’s 2006 six-phase thematic analysis method ref. [32]. Phases to be followed for this study include the following: familiarising oneself with the data, generating initial codes, searching for themes, reviewing themes, defining and naming themes, and, lastly, presenting results as indicated by ref. [33]. The researcher will apply an open coding method to categorise the data. This method involves grouping similar patterns to create meaningful categories. It is anticipated that these categories will yield themes that will help organise the collected data into a coherent format. Additionally, the data collected will be analysed and interpreted, considering the researchers’ knowledge, insights, biases, and experiences related to the study topic ref. [33].

### 2.3. Ethical Considerations

The study proposal was presented at the University of Venda, Department of Advanced Nursing Science, and then submitted to the Faculty of Health Sciences for quality purposes. Thereafter, the proposal was submitted to the University of Venda Higher Degree Research Committee for approval and the issuance of an ethical clearance certificate. Ethical clearance was granted in June 2025. An ethical clearance certificate was submitted to FSDoH (Bophelo House) and the selected public hospital for permission to conduct the study. Communication was made via the Free State Department of Health to Mangaung District, the executive and Head of Nursing in the selected public hospital, and lastly, with participants to sign an informed consent pertaining to the study, for the participants who agreed to participate in the study. The ethical principles will be adhered to.

Anonymity—refers to the manner in which participants will not be identifiable when the collected data is analysed ref. [34]. The researcher will allocate participant numbers instead of addressing them with names and will also not collect information that can be easily linked to participants. The participants will be informed that their identities will not feature anywhere in this study, and they will be allocated numbers instead.

Confidentiality—interviews will be held in a secure, private place, and the data collected will be kept anonymous and will be made available only to those involved in this study. The researcher will reassure the participants that the information shared during the study will be used for the sole purpose of this study, and the collected data will be stored electronically with password-bound access. Participants will be reminded throughout the study that sensitive information can be withheld ref. [27].

Autonomy—pertains to the notion that every rational person has the right to make their own decisions and moral grounds ref. [35]. For this study, participants will be allowed to voluntarily participate in the study, and they will be reminded throughout the study that they can withdraw at any given time without any explanation.

Informed consent—it is viewed as an agreement between the participants and the researcher, bearing in mind that the participants are aware of the risks involved and agree to participate without being coerced ref. [27]. Participants will be made aware that participation in the study is voluntary and that they can withdraw at any time. Participants will also be informed that there will be no monetary incentives for participating in this study, nor will they bear any ownership of the study. All the necessary information pertaining to the study will be explained thoroughly to participants. Participants will then be encouraged to ask questions to clear uncertainties about the study. Participants will be expected to sign an informed consent form as proof of agreement to participate in the study.

### 2.4. Plan for Dissemination

The study findings will be communicated through presentations in nursing webinars, conferences, and publications in accredited nursing journals, articles, and newspapers. An electronic document of the study will be sent to the university library. The study report will be sent and presented to the selected public hospitals’ management and participants.

## 3. Discussion

Ref. [36] defined leadership succession planning as a strategy that is implemented to recognise and prepare potential leaders for management positions within the organisation. Nursing educators can benefit from the study by integrating its findings into curriculum development. By emphasising the importance of leadership skills and providing practical leadership training, educators can better prepare nursing students for future managerial roles. Healthcare administrators can apply the study’s insights to enhance organisational practices that support leadership development. This includes creating a more conducive environment for mentorship, fostering an organisational culture that values leadership growth, and implementing programmes that facilitate the transition from clinical roles to managerial positions. The study’s findings, although centred on a specific local context, could have wider significance for leadership succession planning in nursing worldwide, providing valuable lessons that can be utilised across different healthcare environments.

It will offer valuable insights into the unique challenges and opportunities associated with leadership succession planning in the Mangaung District, considering local healthcare dynamics that may differ from those in other regions of South Africa and around the world. The research will investigate methods to boost engagement and commitment among nursing staff toward leadership positions, potentially resulting in higher job satisfaction and improved retention rates. Ref. [37] indicates that the study’s findings can guide the DoH in creating and refining policies that support leadership development within the nursing workforce.

## 4. Conclusions

Leadership succession planning has become a pressing issue for professional nurses in the Mangaung District, Free State, as a significant number of seasoned nurses approach retirement. This trend poses the risk of creating a substantial gap in experienced leadership. By tackling the obstacles that hinder effective succession planning, healthcare organizations can ease the impact of leadership transitions and secure a continuous supply of capable, well-prepared nursing leaders for the district.

## Data Availability

No new data were created or analysed in this study. Data sharing is not applicable to this article.

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
