# Peer review of "Leadership Succession Planning for Professional Nurses in a Selected Public Hospital in Mangaung District, Free State Province: A Study Protocol"

_healthcare, 2025, doi:10.3390/healthcare13243313_

Round 1

Reviewer 1 Report

Comments and Suggestions for Authors

Leadership Succession Planning for Professional Nurses in 2 Selected Public Hospitals of Mangaung District, Free State 3 Province: Study Protocol

Dear Authors,

Thanks for giving me the chance to review this interesting study protocol. I hope my feedback is helpful for you to enhance the study. Leadership succession planning for nurses in South African public hospitals is highly relevant given the aging workforce and leadership shortages. The study addresses a documented gap in the literature. The aim (to determine leadership succession planning for professional nurses) and the objectives (exploring and recommending strategies) are well defined. The methodology (a qualitative, explorative, descriptive design) is suitable for investigating lived experiences and perceptions. The ethical approval has been obtained, and issues of consent, confidentiality, and autonomy are well described. The protocol clearly describes the setting (Mangaung District hospitals), inclusion criteria, and rationale for participant selection. However, some parts of this study needs some improvement as the following.

There’s a lack of theoretical framework. The study does not ground itself in a leadership or succession planning theory/model. This may weaken the analytical lens and the interpretive depth of findings.

Sections of the introduction and rationale repeat the same arguments (aging workforce, shortage of nurse managers). As this is a study protocol, the narrative could be more concise and better synthesized.

Purposive sampling of 30 participants is reasonable, but the justification for “at least 7 years of clinical experience” is not well explained, which may exclude valuable perspectives (e.g., younger nurses with leadership aspirations).

The central research question is presented, but probing questions or interview guides are not outlined. This leaves uncertainty about the depth of data that will be collected. You had better explain this part under the data collection procedure.

The qualitative rigor elements (credibility, dependability, transferability, confirmability) are listed, they are kind of generic and not tailored to this specific study.

Comments on the Quality of English Language

Generally, there are numerous grammatical errors, awkward phrasing, and inconsistent referencing reduce the professional quality of the manuscript. It would require significant language and style editing before publication.

Author Response

Comment 1: There’s a lack of a theoretical framework. The study does not ground itself in a leadership or succession planning theory/model. This may weaken the analytical lens and the interpretive depth of findings.

Response 1: The Transformational theory model was added. Thank you for pointing this out. We agree with this comments. Therefore, the Transformational theory model was added, page 3, paragraph 1, and lines 94-110

Comments 2: Sections of the introduction and rationale repeat the same arguments (aging workforce, shortage of nurse managers). As this is a study protocol, the narrative could be more concise and better synthesized.

Response 2: The introduction and the rationale arguments are no longer the same. Thank you for pointing this out. We agree with this comments. Therefore,  repetition has been eliminated from the rational, page 4, paragraph 2, and lines 133-150.

Comments 3: Purposive sampling of 30 participants is reasonable, but the justification for “at least 7 years of clinical experience” is not well explained, which may exclude valuable perspectives (e.g., younger nurses with leadership aspirations).

Response 3:  More than 5 years of clinical experience. Thank you for pointing this out. We agree with this comments. Therefore, we have changed “at least 7 years of clinical experience to more than 5 years of clinical experience, page 6, paragraphs 2, and lines 220.

Comments 4: The central research question is presented, but probing questions or interview guides are not outlined. This leaves uncertainty about the depth of data that will be collected. You had better explain this part under the data collection procedure.

Response 4: An Unstructured interview, with one central question, will be used; however,  probing questions will be based on the participants' responses. Thank you for pointing this out. We agree with this comments. Therefore, we have added the probing information, page 7, paragraph 2, and lines 281-283.

Comments 5: The qualitative rigor elements (credibility, dependability, transferability, confirmability) are listed, they are kind of generic and not tailored to this specific study.

Response 5: The qualitative rigor elements are now tailored to the study.  Thank you for pointing this out. We agree with this comments. Therefore, we have added the qualitative rigor elements information, pages 6-7, paragraphs 5-7, and lines 249-276.

Reviewer 2 Report

Comments and Suggestions for Authors

Thank you for submitting this study protocol, "Leadership Succession Planning for Professional Nurses in 2 Selected Public Hospitals of Mangaung District, Free State Province: Study Protocol." This is a critically important topic. The manuscript demonstrates a clear purpose and lays out a thoughtful methodological plan. Specific strengths and areas for improvement are listed below:

Strengths

The manuscript addresses an important gap in the South African healthcare system, where leadership succession planning in the public sector is often neglected.

The introduction includes global and regional comparisons, which helps place the problem in a broader context.

The methodology is generally sound, with a clear qualitative approach, purposeful sampling, and detailed inclusion/exclusion criteria.

Ethical considerations and reliability measures are carefully described, enhancing the rigor of the protocol.

It is well aware of its reach and impact, with potential value for both policy and practice.

Areas for Improvement

1- Introduction and Literature References

Although the introduction provides extensive references, it is somewhat long and repetitive. It is recommended that the gap statement be improved to clarify what is known, what is missing, and the relevance of this study in the context of public hospitals in South Africa.

Some references are not well-integrated; rather than simply citing them, it is advisable to explain their relevance to the research problem.

2- Methodology

The requirement that participants have at least 7 years of clinical experience can be justified. Why not 5 years or more, 10 years or more, or another criterion? Provide justification for this specific requirement.

The guiding interview question could be improved. For example, "How do you view leadership succession planning in your hospital, and what challenges or opportunities do you see in implementing it?" This would encourage richer answers.

Consider whether the expected sample size of 30 participants is realistic, as qualitative studies often reach saturation with fewer interviews.

3- Clarity and Structure

The writing is generally clear, but it would be useful to improve its grammar to improve brevity and eliminate repetition.

Some sentences are too long or imprecisely worded (for example, "It was noted that Saudi Arabia subsequently experienced an increase in its population..."). Short, direct sentences improve readability.

Avoid excessive transitional phrases such as "according to," "it was found that," and "it was noted that."

4- Discussion

The discussion section mostly restates background information rather than outlining the expected contributions of this study. It is advisable to focus on the uniqueness of this protocol and the new insights it can offer compared to existing studies in South Africa and globally.

5- Presentation

Currently, the manuscript does not include any tables or figures. While a summary table of the methodology (study design, location, population, samples, data collection, and analysis) is not essential for the protocol, it would enhance clarity.

Comments on the Quality of English Language

Language

The English language is easy to understand, but could be improved to make it more fluent and professional. I recommend careful review by a native or advanced English-speaking editor.

Author Response

Comments 1: Although the introduction provides extensive references, it is somewhat long and repetitive. It is recommended that the gap statement be improved to clarify what is known, what is missing, and the relevance of this study in the context of public hospitals in South Africa.

Response 1: Gap statement/ problem statement improved. Thank you for pointing this out. We agree with this comment. Therefore, we have adjusted and added suggested information, pages 3-4, paragraph 2, and lines 111-132.

Comments 2 Some references are not well-integrated; rather than simply citing them, it is advisable to explain their relevance to the research problem.

Response 2: References relevance. Thank you for pointing this out. We agree with this comment. Therefore, we have integrated the references to the research problem on pages 1-3, paragraphs 1-5, and lines 31-92

Comments 3: The requirement that participants have at least 7 years of clinical experience can be justified. Why not 5 years or more, 10 years or more, or another criterion? Provide justification for this specific requirement

Response 3:  More than 5 years of clinical experience. More than 5 years of clinical experience. Thank you for pointing this out. We agree with this comment. Therefore, we have changed at least 7 years of clinical experience to more than 5 years of clinical experience, page 6, paragraphs 3, and lines 227.

Comments 4: The guiding interview question could be improved. For example, "How do you view leadership succession planning in your hospital, and what challenges or opportunities do you see in implementing it?" This would encourage richer answers.

Response 4: The guiding interview question was improved. Thank you for pointing this out. We agree with this comment. Therefore, we have improved. Page 7, paragraph 2, line 281-283.

Comments 5: Consider whether the expected sample size of 30 participants is realistic, as qualitative studies often reach saturation with fewer interviews.

Response 5: The sample size of 30 participants is an estimation. Thank you for pointing this out. We agree with this comment. Therefore, we have indicated in the sample size, page 6, paragraph 2, and line 219.

Comments 6 The writing is generally clear, but it would be useful to improve its grammar to improve brevity and eliminate repetition.

Response 6: Manuscript will be edited. Thank you for pointing this out. We agree with this comment. Therefore, we have eliminated repetition, and the manuscript was edited to eliminate repetition to improve the quality.

Comments 7. Some sentences are too long or imprecisely worded (for example, "It was noted that Saudi Arabia subsequently experienced an increase in its population..."). Short, direct sentences improve readability.

Response 7: Some sentences are no longer too long or imprecisely worded. Thank you for pointing this out. We agree with this comment. Therefore, we have made the sentence short on page 2,  paragraph 2, and lines 49-50.

Comments 8: Avoid excessive transitional phrases such as "according to," "it was found that," and "it was noted that."

Response 8: Excessive transitional phrases were avoided. Thank you for pointing this out. We agree with this comment. Therefore, we have implemented the comments throughout the document.

Comments 9: The discussion section mostly restates background information rather than outlining the expected contributions of this study. It is advisable to focus on the uniqueness of this protocol and the new insights it can offer compared to existing studies in South Africa and globally.

Response 9. The discussion section outlines the expected contributions of this study.   Thank you for pointing this out. We agree with this comment. Therefore, we have outlined the expected contributions of this study, page 9, paragraph 3, and line 382-401.

Comments 10: Currently, the manuscript does not include any tables or figures. While a summary table of the methodology (study design, location, population, samples, data collection, and analysis) is not essential for the protocol, it would enhance clarity.

Response 10: Tables and figures not included. Thank you for pointing this out. We agree with this comment. Therefore, Transformational theory model was added, page 3, paragraph 1, and lines 108-109.

Comments 11: The English language is easy to understand, but could be improved to make it more fluent and professional. I recommend careful review by a native or advanced English-speaking editor.

Response 11: Manuscript will be edited. Thank you for pointing this out. We agree with this comment. Therefore, the manuscript was edited.

Reviewer 3 Report

Comments and Suggestions for Authors

The authors describe a study protocol that uses a qualitative design to describe the current state of leadership succession planning and provide recommendations based on unstructured interviews with experienced professional nurses in a public hospital in South Africa. The study had the potential to contribute to our understanding of leadership succession planning from the perspective of nurses who might be considering future leadership roles. The paper would be of interest to a wide audience of researchers, clinicians and administrators around the globe given the continued struggle to identify, motivate and develop nurses for current and future leadership roles. The basic components of a study protocol have been outlined the in the paper but further elaboration and editing are required to enhance the quality of the paper. There are no critical flaws, the comments are minor in nature and are intended enhance the clarity of the study protocol.  Editorial suggestions are also provided for your consideration.

Abstract

-page 1, lines 11-16. The first few sentences have an awkward structure. Suggested edits:

- “There is a significant demand for  nurse managers in Free State Province and therefore leadership succession planning is important.” (or something similar to tie these 2 concepts together)

” The re-advertising of unfilled leadership roles, the projected volume of nurse managers who will be retiring and the number of professional nurses opting for better international opportunities indicate the need for effective succession planning.” (or something similar to connect these ideas)

“The purpose of the study is to determine the current state and recommendations for  leadership succession planning for professional nurses in selected public hospitals in Mangaung District, Free State Province of South Africa.” (the purpose/aim needs to be consistent with the body of the paper)

Introduction

-page 1, line 33  -“ [3] states that it is acknowledged as a better technique in employment across the management board.” This seems to be similar to the previous statement. Suggest adding the citation #3 to the prior sentence e.g,. [2,3].

-line 37 (and several other examples throughout the paper) –Using a reference number instead of the author’s name doesn’t flow in this sentence. The sentences could be rephrased to avoid this issue e.g., for citation 4, suggest the following edit “Globally, factors such as a lack of skilled human resources and limited time frames to find suitable candidates are among the factors that have given rise to succession planning in some institutions [4].

-page 2, line 43 – minor edit “due to failure of not understanding the term succession planning”

-line 47 – It is not clear what is meant by the phrase “take their positions before their time”. Do you mean the new leader is not prepared enough to take on the role or that the new leader is wanting the manager’s role before the manager is ready to leave the position?

-line 61 – Suggested “In addition” instead of “Besides”.

-line 65-68 – Awkward sentence, suggested edit “In Kenya, as per The Ministry of Health’s  National Nursing and Midwifery Policy (2022-2032), recommendations were made regarding strengthening management and leadership practices, the importance of nursing leadership, and the need for improvement of and development of nurse leaders” [12]. (or something similar)

-line 69-70 – The next sentence is unclear regarding gaps found in the programmes.

-line 79 – minor edit “Succession planning is practiced more in private institutions…”

-line 83-85- Awkward sentence. Suggested edit “ Factors that hinder leadership succession planning in the SA state-owned entities include lack of support from the organisation, supervisory (word missing, what type of supervision) and need for performance appraisals [17]. (refer back to the source to capture the exact factors listed)

-line 87 – Did you mean ‘leadership positions’  among hospitals?

-line 90 – you mention ‘certain barriers’ are in place. It would be useful to mention a few of these barriers.

-page 3, line 93-94 – The study aim is not clear or precise. The study objectives on page 4 are much more specific. This is an issue as well on line 115.

-line 98 – Suggested edit “The shortage of nurse managers has resulted in increases in their workload as managers are often responsible for two units.” (so something similar). Using the word ‘this’ is to vague.

-line 100-103 – Awkward sentence, suggested edit “Moreover, leadership succession planning is needed so that leaders can address the existing challenges of an aging workforce that is close to retirement; limited healthcare resources, equipment, and personnel; and the need for a sustainable infrastructure.”

-line 109 – do you mean “limited development opportunities”?

-line 126 – “Thus, the researcher found it necessary to conduct a study based on a nursing perspective.” This statement requires further elaboration. What is the value of the nursing perspective?

Methods

-lines 136-38 – This is redundant as it has bee stated several times previously. The statement of study objectives is very clear and useful. That said, in the paragraphs that follow on this page, you also mention that you want to “articulate the significance of leadership succession planning’ and ‘describe the importance of effective leadership succession planning’. These should be reflected in the study objectives as well.

-page 4 -the explanation of explorative and descriptive designs seems to read like a textbook. It would be more clear to differentiate these 2 types of designs from each other and then indicate how the methods flow from each of these designs.

-Study setting – Some of the details have already been mentioned earlier in the paper. The geographic coordinates and weather patterns are not needed.

-page 5 – sampling of the facilities – On the previous page, you noted that a public hospital has been selected. It is not clear what is meant by the ‘facilities’ as you describe it in this section.

-sample size – Suggest deleting the last sentence as these ideas have been stated multiple times before.

-line 226-27 – suggest deleting the last sentence as it is self evident that nurses with 7 years of experience will be old enough to give consent.

-Page 6, lines 237-239 – Suggest deleting these sentences as they are redundant (found earlier in the paper): “Participants will be selected through a non-probability purposive sampling.” and “The data collected will be analysed through Braun and Clarke’s 2006 thematic analysis method [28].”

-line 241 – It would be unusual to change the research questions if problems were experienced. Usually the interview questions would be revised. A small but important detail, you have research objectives but no research questions listed in this protocol.

I suggest that the use of a semi-structured interview might be more focused and efficient. If you stay with an unstructured interview, further support for this decision might be needed.

-lines 252-255 – The method by which you intend to ensure credibility needs further explanation i.e., how will you engage in peer debriefing, check externally and evaluate the references.

-lines258-62 – Dependability also needs further explanation to understand the steps you will be taking.

-lines 264-266 – suggest deleting the first sentence in this paragraph as it is unclear. The second sentence is adequate.

-line 271 – What will the researcher do to ensure that they are not ‘biased’ through the interview and analysis stages?

-page 7, line 312 – How will the identity of the participants be kept confidential if the nurse manager facilitates their availability?

-line 323 – This is the first time that a ‘supervisor’ has been mentioned. A brief explanation would be useful here.

-page 8, line 325 – Where will the locked box be stored?

-lines 327-29 – Suggest deleting the first 2 sentences as this is basic knowledge for qualitative studies.

-line 365 – The use of a password protected file storage supports confidentiality. The use of  pseudonyms support anonymity. Earlier you stated you would use coded numbers rather than pseudonyms. Select the method you will use and ensure consistency throughout the paper.

-page 9, lines 381+86 – This paragraph does not provide any new information so it could be deleted.

-pages 5-7 -It is convention to avoid using bullet points. I think if the bullet points were removed and the use of subheading was more conservative, the details on these pages would flow better and be more concise.

Author Response

Comments 1 page 1, lines 11-16. The first few sentences have an awkward structure. Suggested edits:

Response 1: A concept to join the sentence was inserted. A concept to join the sentence was inserted. Thank you for pointing this out. We agree with this comment. Therefore, we have a concept to join the sentence was inserted, Page 1, paragraph 1, and line 11-17

Comments 2.  The re-advertising of unfilled leadership roles, the projected volume of nurse managers who will be retiring and the number of professional nurses opting for better international opportunities indicate the need for effective succession planning.” (or something similar to connect these ideas).

Response 2: A concept to join the sentence was inserted. Thank you for pointing this out. We agree with this comment. Therefore, we have a concept to join the sentence was inserted, Page 1, paragraph 1, and line 11-17

Comments 3. “The purpose of the study is to determine the current state and recommendations for  leadership succession planning for professional nurses in selected public hospitals in Mangaung District, Free State Province of South Africa.” (the purpose/aim needs to be consistent with the body of the paper).

Response 3: The aim/purpose is consistent throughout the document. The aim/purpose is consistent throughout the document. Thank you for pointing this out. We agree with this comment. Therefore, the aim/purpose is consistent throughout the document. Page 1 and 2-3, paragraph 1 and lines 15-17 and 91-93.

Comments 4: -page 1, line 33  -“ [3] states that it is acknowledged as a better technique in employment across the management board.” This seems to be similar to the previous statement. Suggest adding the citation #3 to the prior sentence e.g,. [2,3].

Response 4: Statement was edited. The statement was edited. Thank you for pointing this out. We agree with this comment. Therefore, we have edited and deleted the statement. Page 1, paragraph 1, and line 32-34.

Comments 5: -line 37 (and several other examples throughout the paper) –Using a reference number instead of the author’s name doesn’t flow in this sentence. The sentences could be rephrased to avoid this issue e.g., for citation 4, suggest the following edit: “Globally, factors such as a lack of skilled human resources and limited time frames to find suitable candidates are among the factors that have given rise to succession planning in some institutions [4].

Response 5: The implementation of the comment is done. Thank you for pointing this out. We agree with this comment. Therefore, we have the implementation of the comment done. Page 1, paragraph 1, and lines 36-38.

Comments 6: -page 2, line 43 – minor edit “due to failure of not understanding the term succession planning”

Response 6: Editing done. Thank you for pointing this out. We agree with this comment. Therefore, we have edited the statement, Page 2, paragraph 1, and line 42.

Comments 7: -line 47 – It is not clear what is meant by the phrase “take their positions before their time”. Do you mean the new leader is not prepared enough to take on the role or that the new leader is wanting the manager’s role before the manager is ready to leave the position?.

Response 7: Statement was edited. Thank you for pointing this out. We agree with this comment. Therefore, we have edited the statement. Page 2, paragraph 1, and lines 45-46

Comments 8: -line 61 – Suggested “In addition” instead of “Besides”.

Response 8: In addition was added. Thank you for pointing this out. We agree with this comment. Therefore, we have added, in addition, page 2, paragraph 3, and line 60.

Comments 9: -line 65-68 – Awkward sentence, suggested edit “In Kenya, as per The Ministry of Health’s National Nursing and Midwifery Policy (2022-2032), recommendations were made regarding strengthening management and leadership practices, the importance of nursing leadership, and the need for improvement of and development of nurse leaders” [12]. (or something similar)

Response 9: Suggestion implemented. Thank you for pointing this out. We agree with this comment. Therefore, we have implemented the suggestion. Page 2,  Paragraph 3 and lines 64-68.

Comments 10: -line 69-70 – The next sentence is unclear regarding gaps found in the programmes.

Response 10: Unclear sentence removed. Unclear sentence removed. Thank you for pointing this out. We agree with this comment. Therefore, we have removed the sentence.

Comments 11-line 79 – minor edit “Succession planning is practiced more in private institutions…”

Response 11: Sentence edited. Thank you for pointing this out. We agree with this comment. Therefore, we have edited sentences. Page 2, paragraph 4, and lines 76-77

Comments 12-line 83-85- Awkward sentence. Suggested edit “ Factors that hinder leadership succession planning in the SA state-owned entities include lack of support from the organisation, supervisory (word missing, what type of supervision) and need for performance appraisals [17].

Response 12: Sentence edited. Thank you for pointing this out. We agree with this comment. We agree with this comment. Therefore, we have edited the sentence. Page 2, paragraph 4, and lines 80-82.

Comments 13: – Did you mean ‘leadership positions’ among hospitals?

Response 13:  Edited. Thank you for pointing this out. We agree with this comment. Therefore, we have added the positions. Page 2, paragraph 5, and line 84.

Comments 14: -line 90 – you mention ‘certain barriers’ are in place. It would be useful to mention a few of these barriers.

Response 14:  Lack of succession planning, mentorship, and failure to recognize postgraduate qualifications. Thank you for pointing this out. We agree with this comment. Therefore, we have added the barriers. Page 2, paragraph 5, and lines 87-89.

Comments 15: page 3, line 93-94 – The study aim is not clear or precise. The study objectives on page 4 are much more specific. This is an issue as well on line 115.

Response 15:  The aim is aligned throughout the document. Thank you for pointing this out. We agree with this comment. Therefore, we have aligned the aim, page 3 and line 91-93.

Comments 16: line 98 – Suggested edit “The shortage of nurse managers has resulted in increases in their workload as managers are often responsible for two units.” (something similar). Using the word ‘this’ is too vague.

Response 16: Statement is edited. Thank you for pointing this out. We agree with this comment. Therefore, we have edited the statement, page 3, paragraph 2, and lines 114-115.

Comments 17: -line 100-103 – Awkward sentence, suggested edit “Moreover, leadership succession planning is needed so that leaders can address the existing challenges of an aging workforce that is close to retirement; limited healthcare resources, equipment, and personnel; and the need for a sustainable infrastructure.”

Response 17: Statement is edited. Thank you for pointing this out. We agree with this comment. Statement is edited. Page 3, Paragraph 3, Line 117-119.

Comments 18: line 109 – do you mean “limited development opportunities”?

Response 18: Statement is edited. Thank you for pointing this out. We agree with this comment. Therefore, we have edited the statement. Pages 3-4, paragraph 1, and lines 124-125.

Comments 19 line 126 – “Thus, the researcher found it necessary to conduct a study based on a nursing perspective.” This statement requires further elaboration. What is the value of the nursing perspective?

Response 19: Sentence deleted. Thank you for pointing this out. We agree with this comment. Therefore, we have deleted the sentence.

Comments 20: -lines 136-38 – This is redundant as it has been stated several times previously. The statement of study objectives is very clear and useful. That said, in the paragraphs that follow on this page, you also mention that you want to “articulate the significance of leadership succession planning’ and ‘describe the importance of effective leadership succession planning’. These should be reflected in the study objectives as well.

Response 20: Removed. Thank you for pointing this out. We agree with this comments. Therefore, we have removed articulate the significance of leadership succession planning’ and ‘describe the importance of effective leadership succession planning.

Comments 21: -page 4 -the explanation of explorative and descriptive designs seems to read like a textbook. It would be more clear to differentiate these 2 types of designs from each other and then indicate how the methods flow from each of these designs.

Response 21: Explorative and descriptive designs differentiated, defined, and aligned to the study. Thank you for pointing this out. We agree with this comment. Therefore, we have implemented the comment. Page 4-5, paragraph 6 and 1, and lines 165-180.

Comments 22: -Study setting – Some of the details have already been mentioned earlier in the paper. The geographic coordinates and weather patterns are not needed.

Response 22: The geographic coordinates and weather patterns are removed. Thank you for pointing this out. We agree with this comment. Therefore, we have removed the sentence.

Comments 23: -page 5 – sampling of the facilities – On the previous page, you noted that a public hospital has been selected. It is not clear what is meant by the ‘facilities’ as you describe it in this section.

Response 23: Facility added. Thank you for pointing this out. We agree with this comment. Therefore, we have replaced facilities with facility. Therefore, we have replaced facilities with facility. Page 5, paragraph 4, and line 205.

Comments 24: -sample size – Suggest deleting the last sentence as these ideas have been stated multiple times before.

Response 24: Sentence deleted. Thank you for pointing this out. We agree with this comment. Therefore, we have deleted the sentence.

Comments 25: -line 226-27 – suggest deleting the last sentence as it is self-evident that nurses with 7 years of experience will be old enough to give consent.

Response 25: Sentence deleted. Thank you for pointing this out. We agree with this comment. Therefore, we have deleted the sentence.

Comments 26-Page 6, lines 237-239 – Suggest deleting these sentences as they are redundant (found earlier in the paper): “Participants will be selected through a non-probability purposive sampling.” and “The data collected will be analysed through Braun and Clarke’s 2006 thematic analysis method [28].”

Response 26: Sentence deleted. Thank you for pointing this out. We agree with this comment. Therefore, we have deleted the sentence.

Comments 27: -line 241 – It would be unusual to change the research questions if problems were experienced. Usually, the interview questions would be revised. A small but important detail, you have research objectives, but no research questions listed in this protocol. I suggest that the use of a semi-structured interview might be more focused and efficient. If you stay with an unstructured interview, further support for this decision might be needed.

Response 27: An Unstructured interview, with one central question, will be used; however,  probing questions will be based on the participants' responses. Thank you for pointing this out. We agree with this comment. Therefore, we have supported the statement-An Unstructured interview, with one central question, will be used; however, probing questions will be based on the participants' responses. Page 7, paragraph 2, and lines 279-290.

Comments 28-lines 252-255 – The method by which you intend to ensure credibility needs further explanation i.e., how will you engage in peer debriefing, check externally and evaluate the references.

Response 28:  Done. Done. Thank you for pointing this out. We agree with these comments. Therefore, we have explained credibility. Page 6, paragraph 6, and lines 249-254.

Comments 29: lines 258-62 – Dependability also needs further explanation to understand the steps you will be taking

Response 29:  Done. Thank you for pointing this out. We agree with this comment. Therefore, we have explained dependability. Page 6, paragraph 7, and lines 255-261.

Comments 30 -lines 264-266 – suggest deleting the first sentence in this paragraph as it is unclear. The second sentence is adequate.

Response 30: Sentence deleted. Thank you for pointing this out. We agree with this comment. Therefore, we have deleted the sentence.

Comments 31: -line 271 – What will the researcher do to ensure that they are not ‘biased’ through the interview and analysis stages?

Response 31: Statement is edited. Thank you for pointing this out. We agree with this comment. Therefore, we have edited the statement. Page 7, paragraph 1, and line 269-276.

Comments 32: -line 323 – This is the first time that a ‘supervisor’ has been mentioned. A brief explanation would be useful here.

Response 32: Statement is edited. Thank you for pointing this out. We agree with this comment. Therefore, we have added the research supervisor. Page 8, paragraph 2, and lines 322-323.

Comments 33: -page 8, line 325 – Where will the locked box be stored?

Response 33: Statement is edited. Thank you for pointing this out. We agree with this comment. Therefore, we have added a lockable cupboard. Page 8, paragraph 2, and lines 324-325.

Comments 34: lines 327-29 – Suggest deleting the first 2 sentences as this is basic knowledge for qualitative studies.

Response 34: Sentence deleted. Thank you for pointing this out. We agree with this comment. Therefore, we have deleted the sentence.

Comments 35: -line 365 – The use of a password-protected file storage supports confidentiality. The use of pseudonyms support anonymity. Earlier, you stated you would use coded numbers rather than pseudonyms. Select the method you will use and ensure consistency throughout the paper.

Response 35: Pseudonym's name deleted. Thank you for pointing this out. We agree with this comment. Therefore, we have used a password-protected file storage supports.  Page 8 and 9, paragraph 5 and 1, and lines 355-356 and 359-361.

Comments 36: -page 9, lines 381+86 – This paragraph does not provide any new information so it could be deleted.

Response 36: Sentence deleted. Thank you for pointing this out. We agree with this comment. Therefore, we have deleted the sentence.

Comments 37: -pages 5-7 -It is conventional to avoid using bullet points. I think if the bullet points were removed and the use of subheadings was more conservative, the details on these pages would flow better and be more concise.

Response 37: Done. Thank you for pointing this out. We agree with this comment. Therefore, we have removed bullet points, pages 5-7, and replaced them with numbers.

Round 2

Reviewer 1 Report

Comments and Suggestions for Authors

Dear Authors, 

Thanks for responding to my comments. I have no further feedback for you. 

Reviewer 2 Report

Comments and Suggestions for Authors

Thank you, the manuscript is well-structured